# Dose Escalation of Oropharyngeal Cancer: Long-Time Follow-Up and Side Effects

**DOI:** 10.3390/cancers15092580

**Published:** 2023-04-30

**Authors:** Anna Embring, Eva Onjukka, Claes Mercke, Ingmar Lax, Anders Berglund, Signe Friesland

**Affiliations:** 1Department of Oncology, Karolinska University Hospital, 17176 Stockholm, Sweden; 2Karolinska Institute, Department of Oncology-Pathology, 17176 Stockholm, Swedeningmar.lax@ki.se (I.L.); 3Department of Medical Radiation Physics and Nuclear Medicine, Karolinska University Hospital, 17176 Stockholm, Sweden; 4Epistat Epidemiology and Statistics Consulting, 75655 Uppsala, Sweden

**Keywords:** radiotherapy, dose escalation, oropharyngeal cancer, head and neck cancer, side effects, osteoradionecrosis, dysphagia

## Abstract

**Simple Summary:**

In head and neck cancer, local recurrences are unfortunately common. A more intense local treatment, such as an increased radiation dose to the primary tumour, could potentially prevent some of these recurrencies. Previous studies have not shown an increased risk of side effects with a higher radiotherapy dose. However, the follow-up time in these studies is sometimes limited, and late side effects may then be missed. In our study with a relatively long patient follow-up, we compared survival and side effects in two groups with 215 patients in each group, where one group was treated with standard dose radiotherapy and the other group was treated with a higher dose of radiotherapy. In this study, we found that certain severe side effects are more common after higher doses of radiotherapy compared to after treatment with standard dose radiotherapy.

**Abstract:**

Previous studies on dose-escalated radiotherapy in head and neck cancer have shown mixed results, and it is not established which patients would benefit from dose escalation. Further, while dose escalation does not appear to increase late toxicity, this needs to be confirmed with longer follow-up. In this study, we analysed treatment outcome and toxicity in 215 patients with oropharyngeal cancer treated with dose-escalated radiotherapy (>72 Gy, EQD2, α/β = 10 Gy, boost by brachytherapy or simultaneous integrated boost) and a matched cohort of 215 patients treated with standard dose external-beam radiotherapy (68 Gy) between 2011 and 2018 at our institution. The 5-year overall survival (OS) was 77.8% (72.4–83.6) and 73.7% (67.8–80.1) in the dose-escalated and standard dose group, respectively (*p* = 0.24). Median follow-up was 78.1 (49.2–98.4) and 60.2 (38.9–89.4) months in the dose-escalated and standard dose groups, respectively. Grade ≥3 osteoradionecrosis (ORN) and late dysphagia were more common in the dose-escalated group compared to the standard dose group, with 19 (8.8%) vs. 4 (1.9%) patients developing grade ≥3 ORN (*p* = 0.001), and 39 (18.1%) vs. 21 (9.8%) patients developing grade ≥3 dysphagia (*p* = 0.01). No predictive factors to help select patients for dose-escalated radiotherapy were found. However, the remarkably good OS in the dose-escalated cohort, despite a predominance of advanced tumour stages, encourages further attempts to identify such factors.

## 1. Introduction

Despite the latest decades of medical and technical advancements, many patients with head and neck cancer (HNC) still succumb to the disease, and the 5-year relative survival is 65–72% [1,2]. The most common pattern of treatment failure is local recurrence, which occurs in 18–50% of patients [3,4,5], and it could therefore be hypothesised that some subgroups of HNC patients would benefit from an intensified local treatment, such as radiation dose escalation. However, treating advanced HNC is a challenge due to the tumour location being close to several structures involved in important functions such as eating, speaking, and breathing, and hence dose escalation requires a fine balancing act between achieving improved local control and managing the increased risk of side effects. Several studies have shown that dose escalation in HNC can be achieved with acceptable side effects [6,7,8,9], but often the follow-up time has been limited. To investigate whether this can be confirmed with a longer follow-up, this study specifically evaluated the late side effects of the skin, mucosa, larynx, salivary glands, mandible/bones, dysphagia, and trismus, and we also evaluated the acute side effects of the skin, mucosa, larynx, and trismus.

It remains unclear which subgroup of patients could benefit from intensified treatment. To our best knowledge, there are no studies showing that an increase in radiation dose will result in an improved local control that also translates into an increase in overall survival (OS) for oropharyngeal cancer patients. Factors that have been shown to have a negative impact on survival in oropharyngeal cancer are human papillomavirus (HPV) negative tumours, advanced T- or N-stage, and number of pack-years of tobacco smoking [10]. It is primarily in this group of poorer prognoses that new strategies are needed to improve treatment outcomes and for whom dose escalation could be a possible way to achieve improved results. This was explored in the ARTSCAN III study, where HNC patients with T3 and T4 tumours were randomised to either dose-escalated or standard dose radiotherapy [7]. However, the investigators did not find any significant difference in local control between the two groups. In 2020, Atwell et al. [8] published a review of 18 studies on dose escalation in locally advanced HNC and found that both survival and toxicities were comparable to previous reports on standard dose treatments. The definition of dose-escalated radiotherapy used by Atwell et al., that is, doses considered above standard dose radiotherapy in the National Comprehensive Cancer Network Guidelines [11] (>72 Gy, EQD2, α/β = 10 Gy), is the same as the definition used in the current study.

The aim of the current study is to compare survival and side effects in patients treated for oropharyngeal cancer with dose-escalated radiotherapy and a matched control group that have received standard dose radiotherapy. Compared to many previous studies, the current study analyses a relatively large cohort with a long follow-up time, and toxicity data were prospectively collected.

## 2. Materials and Methods

### 2.1. Patients

Clinical investigations at diagnosis included a complete medical history, physical examination, and panendoscopy performed by an ear, nose, and throat surgeon, and a diagnostic computed tomography (CT) scan with intravenous contrast. Before start of treatment, all patients were considered at a multidisciplinary tumour board for staging and to decide the most suitable treatment modality. All patients were staged according to the American Joint Committee on Cancer’s (AJCC) Cancer-Staging Manual, 7th edition [12]. The final decision to treat with either standard dose radiotherapy or dose-escalated radiotherapy was made by the treating oncologist based on local guidelines and standard clinical procedures including taking the general condition and preferences of the patient into account and performing an assessment of the intended treatment plan. The guidelines primarily recommended dose escalation for oropharyngeal cancers with advanced stage primary tumours but also optionally for T1 and T2 tumours of the base of tongue as they have poorer prognosis than tonsillar cancer with a similar size primary tumour [13,14]. The size of the primary tumour (GTVT) analysed in this study was defined as the GTVT delineated by the treating oncologist. 

We identified 244 patients consecutively treated with dose escalation and curative intent for oropharyngeal squamous cell carcinoma between 2011 and 2018 at our centre. The definition of dose escalation used in this study is a prescribed dose to the primary tumour exceeding standard treatment using external beam photon therapy as defined in the National Comprehensive Cancer Network (NCCN) guidelines [15], i.e., >72 Gy (EQD2, α/β = 10 Gy). Patients were mainly identified through a local quality registry, but as some patients were not recorded in the registry (for example, due to failing to attend follow-up visits), a manual review of our treatment planning system was performed to find all eligible patients. 

During the same period, 332 patients with oropharyngeal squamous cell carcinoma were treated with 68 Gy in 2 Gy fractions (curative intent) at our centre. These patients were similarly identified through our local quality registry supplemented by a manual search of our treatment planning system. After matching on the tumour stage, concurrent medical treatment, and data source (local quality registry or treatment planning system), 215 patients remained in each treatment group. Thus, a total of 430 patients were included in this comparison of dose-escalated and standard dose radiotherapy.

The study was approved by the National Ethical Review Authority.

### 2.2. Treatment

#### 2.2.1. Standard Dose Radiotherapy

The patients were immobilised during radiotherapy using a thermoplastic 5-point mask and treated in a supine position. For treatment planning a contrast enhanced CT scan with 2–2.5 mm slice thickness was used. The treatment was 68 Gy in 2 Gy fractions to high-risk volumes, which consisted of the gross tumour volume (GTV) with a minimum margin of 10 mm. Elective radiotherapy was routinely applied to levels II-IV bilaterally with either 46 Gy in 2 Gy fractions when a sequential boost was used or 51.68 Gy using 1.52 Gy per fraction (EQD2 = 49.6 Gy (α/β = 10)) when the radiotherapy was delivered using a simultaneous integrated boost (SIB). The planning target volume (PTV) was created by an isotropic 5 mm margin and the standard radiotherapy technique used was volumetric modulated arc therapy (VMAT) and 6 MV photons. The Eclipse treatment planning system was used (Varian, USA).

#### 2.2.2. Dose-Escalated Radiotherapy

The dose-escalated radiotherapy was delivered with either external beam radiotherapy only, using SIB, or external beam radiotherapy with a sequential brachytherapy boost. The typical dose-escalated radiotherapy by SIB was delivered with 74.8 Gy in 2.2 Gy fractions (EQD2 = 76.0 Gy (α/β = 10)) to the primary tumour with a 0–5 mm margin, 68 Gy in 2 Gy fractions to GTV + 10 mm margin, and elective radiotherapy to levels II-IV bilaterally with 51.68 Gy using 1.52 Gy per fraction (EQD2 = 49.6 Gy (α/β = 10)). The PTV was created by an isotropic 5 mm margin for the standard and elective dose volumes, and an isotropic 3 mm margin for the dose-escalated volume. The external beam radiotherapy was planned in Eclipse (Varian, USA). The patients receiving dose-escalated radiotherapy through a brachytherapy boost first received external beam radiotherapy to 68 Gy as described above in the standard dose treatment. Approximately one week after the external beam radiotherapy, they had a brachytherapy boost to the primary tumour with a 5–10 mm safety margin typically using pulsed dose rate (PDR) consisting of 15 fractions, with 0.56–0.66 Gy per fraction (total dose in EQD2 = 75.4–76.8 Gy (α/β = 10). The brachytherapy treatments were planned in Oncentra (Elekta, Sweden). For more detailed information of the dose-escalated radiotherapy, see the description in our earlier study comparing these two different modalities of dose-escalated radiotherapy [16].

#### 2.2.3. Medical Treatment

In both treatment groups, the standard concomitant treatment was either weekly cisplatin, with 40 mg/m^2^ administered intravenously once weekly during radiotherapy (maximum dose of 70 mg) or weekly cetuximab, starting with an intravenous loading dose of 400 mg/m^2^ one week before start of radiotherapy and thereafter 250 mg/m^2^ weekly during radiotherapy. 

### 2.3. Toxicity Outcomes

Toxicity data in this study was acquired from our institutional local quality registry and complemented with a review of medical records in the case of missing data. The local quality registry contains data on treatment and tumour characteristics, as well as prospectively gathered data on side effects on all patients treated for HNC with curative intent at our centre. Data on side effects are recorded at regular follow-up visits every 3 months during the first 2 years and then every 6 months for the following 3 years. Toxicities recorded in the registry are osteoradionecrosis (ORN), dysphagia, xerostomia, and trismus, as well as mucosal, laryngeal, and skin toxicities. Grading of ORN is according to Late effects Normal Tissue Task Force Subjective, Objective, Management, and Analytic (LENT/SOMA) scores [17] and the remaining side effects are graded according to the Radiation Therapy Oncology Group (RTOG) and the European Organization for Research and Treatment of Cancer (EORTC) score [18]. Side effects are considered acute when occurring during radiotherapy or within 90 days of end of radiotherapy. Side effects presenting later than that are considered late side effects. In the current study, side effects were considered severe at grade ≥3. To evaluate the incidence of serious side effects, an interim analysis was conducted in 2015 and no significant increase in serious side effects was found at this point. The closure of database was 17 September 2021.

### 2.4. Statistics

The matched cohort was based on the Propensity Score Matching (PSM) procedures. PSM is a method to minimise selection bias between interventional groups in observational studies, in which the propensity score is the probability of intervention assignment conditional on the baseline characteristics. The propensity scores were developed accounting for tumour stage, concurrent medical treatment, and data source (local quality registry or treatment planning system). All individual propensity scores were calculated through logistic regression models and then a 1:1 nearest neighbour propensity score matching with a calliper size of 0.1 was used.

Clinical characteristics by dose level was presented using descriptive statistics and tested by chi-square tests for categorical data, and t tests for continuous variables. The Kaplan–Meier method was used to estimate OS and progression-free survival (PFS) from the last day of radiotherapy to progression, death, or end of follow up. Overall mortality was then estimated using a multivariable Cox regression model with hazard ratios (HR) and 95% confidence intervals (CI) in which continuous variables were normalised by multiplication with the standard deviation of the parameter of interest in the cohort.

In a subsequent step, the cumulative incidence of a composite outcome was described using the cumulative incidence function. The composite outcome was defined as the first event of adverse events (grade ≥3), progression, or death, whichever comes first.

All tests were 2-sided and statistical significance was considered with a *p*-value less than 0.05. The statistical analyses were performed using R version 3.6.1. (R basis for statistical calculation, Vienna University of Economics and Business, Vienna, Austria).

## 3. Results

Four hundred and thirty patients with oropharyngeal squamous cell carcinoma treated with radiotherapy and curative intent at our centre between 2011 and 2018 were included in the study. The mean age at the start of radiotherapy was 63 years and the majority of patients were male (72%). There was a predominance of HPV-positive tumours (83%), and the most common tumour type was tonsillar cancer (60%) followed by base of tongue cancer (36%). There were 215 patients in each of the two matched treatment groups, comprising patients treated with dose-escalated radiotherapy in one treatment group and standard dose radiotherapy in the other. The patients in the dose-escalated treatment group were slightly younger compared to the standard dose treatment group, with a mean age of 62.2 years compared to 64.5 years, respectively. Base of tongue cancer was more common in the dose-escalated group and so was more advanced tumour stage and T-stage. For details on the clinical characteristics of the total cohort and the two different treatment groups, see Table 1. Median follow-up was 78.1 months (interquartile range 49.2–98.4) in the dose-escalated group and 60.2 months (interquartile range 38.9–89.4) in the standard dose group.

The 5-year OS was 77.8% (95% CI 72.4–83.6) and 73.7% (95% CI 67.8–80.1) in the dose-escalated and standard dose group, respectively (*p* = 0.24) and the 5-year PFS was 74.8% (95% CI 69.1–80.9) and 72.7% (95% CI 66.8–79.1) in the dose-escalated and standard dose group, respectively (*p* = 0.58) (Figure 1). A multivariable analysis showed that HPV-positivity was a positive prognostic factor with a hazard ratio of 0.45 (95% CI 0.30–0.68). Performance status ≥1, current smoking, older age, and larger volume of the primary tumour (GTVT) were found to be negative prognostic factors with hazard ratios of 1.51 (95% CI 1.01–2.24), 2.57 (95% CI 1.57–4.23), and 1.04 (95% CI 1.02–1.06) per year and 1.01 (95% CI 1.00–1.02) per cm^3^, respectively (Table 2). Prescribed dose was not found to be a significant prognostic factor of mortality.

Grade ≥3 ORN and late dysphagia were more common in the dose-escalated group compared to the standard dose group, with 19 (8.8%) vs. 4 (1.9%) patients developing grade ≥3 ORN in the dose-escalated and standard dose group, respectively (*p* = 0.001), and 39 (18.1%) vs. 21 (9.8%) patients developing grade ≥3 late dysphagia in the dose-escalated and standard dose groups, respectively (*p* = 0.01). There were no other significant differences in the investigated grade ≥3 side effects (Table 3). The most common severe acute side effect was mucosal side effects (mucositis grade 3 or ulceration), which was found in 61.2% of the total cohort, with no significant difference between the two groups. Nine patients were thought to have died due to treatment related toxicity: six patients in the dose-escalated group and three patients in the standard dose group. Five patients died of acute treatment toxicity and three patients died of late treatment toxicity. One patient was considered to have died of consequential late effects. This patient had severe acute mucosal and skin toxicity during treatment and 2 months later persistent necrotic areas in the pharynx which led to a massive pharyngeal bleeding (without tumour presence) and the need of an acute tracheostomy that unfortunately perforated the posterior pharyngeal wall and subsequently led to fistulas to the mediastinum. This patient died 3 months later after repeated episodes of infection and breathing difficulties. For further details on the grade 5 toxicities, see Table 4.

A composite outcome analysis showed that 5 years after treatment 61.4% (95% CI 55.1–68.3) and 63.8% (95% CI 57.5–70.8) of the patients in the dose-escalated group and in the standard dose group, respectively, were alive, recurrence free, and had no serious late side effects (*p* = 0.58) (Figure 2a) and there was no significant difference in time to develop severe late side effects in the two groups (Figure 2b).

## 4. Discussion

In this retrospective study of patients with oropharyngeal cancer treated with radiotherapy and curative intent, we compared outcome after treatment with standard dose radiotherapy and dose-escalated radiotherapy and found no significant difference in survival regarding prescribed dose. The OS at 5 years was 77.8% (95% CI 72.4–83.6) and 73.7% (95% CI 67.8–80.1) in the dose-escalated and standard dose group, respectively. Our results compare favourably with the relative 5-year survival in oropharyngeal cancer of 72.8% reported by the National Cancer Institute in the United States of America [19] and the relative 5-year survival of 72% from the Swedish Head and Neck Cancer Registry [20]. Despite the predominance of advanced stage primary tumours in the dose-escalated cohort, we found a remarkably good OS compared to the review by Atwell et al. [8], which reported a 3-year locoregional control in non-nasopharyngeal HNC of 68.2–85.9% and 3-year OS of 48.4–51.9% in patients treated with dose-escalated radiotherapy. The reasons for our superior results are not fully understood, but one contributing factor could be that our study included only oropharyngeal cancer while other studies have also included some patients with other types of HNC with poorer prognoses. Explanations relating to the effectiveness of the treatment and the patient selection could also be considered. Furthermore, the effect of unmeasured confounding factors, such as comorbidities, can lead to bias in treatment effect estimates as OS. However, in line with the conclusions by Atwell et al., who considered the results from dose-escalation consistent with previous reports on standard dose radiotherapy, we also could not demonstrate a significant improvement in our dose-escalated cohort over the standard dose cohort.

With our relatively large cohort of patients treated with dose-escalated radiotherapy and matched patients treated with standard dose radiotherapy, we sought to identify a subgroup of patients who might benefit from dose-escalation radiotherapy to the primary tumour. However, while the current study failed to prove the benefit of dose escalation, it does not follow that such a subgroup does not exist. More advanced T-stage tumours could be expected to benefit more from dose-escalated radiotherapy compared to smaller tumours, as patients with a more advanced T-stage have poorer prognosis than patients with early stage tumours [10,21,22] and a larger tumour burden in general requires higher doses to eradicate all tumour cells than a smaller tumour burden [23]. This hypothesis is supported by the findings in a recently published post-hoc exploratory analysis indicating that large tumours might benefit from intensified radiotherapy [24]. To test this hypothesis, we compared survival outcome according to prescribed dose in subgroups stratified by T-stage. Unfortunately, for large primary tumours, the distribution of negative prognostic factors was strongly skewed towards the control group, making the analysis of the influence of prescribed dose impossible (see Appendix A).

In accordance with previous publications, patients in the current study with HPV-positive tumours had better treatment outcomes than patients with HPV-negative tumours [10]. We also found that smoking at the time of treatment, older age, and worse performance status were negative prognostic factors. This is also in accordance with previously published literature [10,25,26]. In our study, the size of the GTVT was a stronger negative prognostic factor than T-stage. This is concurrent with the findings of Adrian et al. [24] and could be a valuable clinical factor to take in consideration in future studies exploring predictive factors in dose-escalation in HNC.

Earlier published studies on dose escalation in HNC have mainly been designed to show feasibility [6,9,27]. In a phase I study on dose escalation in laryngeal and hypopharyngeal cancer, moderately accelerated chemoradiotherapy was considered safe and feasible [28] and in the long-term follow-up, results showed that dose-escalated radiotherapy achieved higher 5-year local control and survival rates with acceptable late side effects, albeit the study was underpowered to show significant survival differences between the two cohorts [29]. However, when the same dose-escalated regimen was used in the phase III randomised controlled trial, ART DECO, there was no improvement of locoregional control or survival outcomes in the dose-escalated arm [30]. The ART DECO study was closed prematurely as a pre-planned interim analysis indicated futility.

In the previously mentioned review article by Atwell et al. [8], it was concluded that not only the survival rates were similar to historical data from standard dose treatments but also the side effects observed were comparable to previously published results for standard dose radiotherapy. However, the authors point out that the results should be interpreted with caution as most studies were small and had short follow-up time, which could result in underreporting of late side effects; the median follow-up has been 1–3 years [27,31,32,33] or less. Indeed, the current study, with much longer follow-up (median 6.5 years in dose-escalated group), demonstrated some higher rates of severe toxicity. As shown in the study by Baudelet et al., late effects after HNC radiotherapy do not seem to be stable over time and for example the prevalence of grade ≥2 dysphagia increased during the 5- to 8-years of follow-up [34]; thus, a longer follow-up shows a more true picture of the severe side effects after radiotherapy.

In the current study, grade ≥3 dysphagia and ORN were more common in the dose-escalated group than in the standard dose group, with 18.1% of the patients who experienced severe late dysphagia in the dose-escalated group compared to 9.8% in the standard dose group. The level of severe late dysphagia in the dose-escalated group is somewhat higher than earlier published data on patients that have received external beam radiotherapy to standard curative dose, where severe late dysphagia is seen in 5–15% [34,35,36,37,38]. Grade ≥3 ORN in the current study was seen in 8.8% of patients in the dose-escalated group compared to 1.9% in the standard dose group. Moreover, with ORN we see an incidence in the dose-escalated group that is on the higher end of what is seen in previously published data of ORN after external beam radiotherapy to a standard curative dose, where the incidence is 3–8% [38,39,40,41,42,43]. The higher rate of severe side effects among the dose-escalated patients is not entirely unexpected as several previous studies have shown that there is a dose-response relationship in the risk of developing side effects [43,44,45] and that oropharyngeal tumours are located close to the relevant structures correlated to these side effects; that is, the mandible and structures involved in swallowing. Moreover, as argued above, our relatively long follow-up might have contributed to detecting a high rate of late side effects compared to previous studies and, more specifically, the follow-up is longer in the dose-escalated group compared to the standard dose group, which could have contributed to the higher rate of late toxicity in the dose-escalated group compared to the standard dose group. However, the analysis of time to develop severe late side effects (Figure 2b) imply that most of the side effects recorded occurred within 5 years after radiotherapy and therefore the difference in follow-up time may not have had a major impact on the results. The longer follow-up in the dose-escalated group can partially be explained by the fact that dose escalation was more commonly practised earlier in the studied period, and thus more time has passed since many of these patients were treated. Another contributing factor to the seemingly high rates of side effects in the dose-escalated group could be the imbalance in T-stage distribution with more patients having advanced T-stages in this group, as Machtay et al. have shown that advanced T-stage is a risk factor for developing severe late toxicity after chemoradiation for locally advanced HNC [46].

There were no statistically significant differences in the incidence of late mucosal ulcers and soft tissue necrosis between the treatment groups, and the incidence was lower (2.3% in dose-escalated group) than previously described in patients treated with dose-escalated radiotherapy where 20–23% grade 4 mucosal ulcers have been reported [47,48]. The reason for this is not clear, but one contributing factor could be the relatively moderate dose escalation in our study. In the current study, we found no statistically significant differences in grade ≥3 trismus or xerostomia between the two treatment groups, and the findings were on par with earlier publications, with severe trismus in 3–18% [38,49], grade ≥3 xerostomia in 1–12% [34,37,38,50], and patient-reported moderate to severe xerostomia in 39% [51]. Regarding severe acute mucosal side effects (grade 3 mucositis and ulceration), we found no significant difference according to dose level, and the reported incidence in the total cohort was 61.2%, which is similar to previously published material where severe acute mucosal side effects is reported in 30–72% [7,9,10]. Side effects of grade 2 or less were not analysed in the current study.

Nine patients (2%) were thought to have died due to treatment-related toxicity. This is on par with earlier published data on treatment related deaths in patients after radiotherapy for oropharyngeal cancer of 0–4% [52,53,54,55,56,57,58,59]. The majority (89%) of these patients had a good performance status (PS 0 or 1) at the start of radiotherapy and most of them (78%) were clinical tumour stage IVA. The causes of death are heterogenous, with acute, late, and consequential late side effects, and with so few cases it is difficult to draw conclusions from any statistical analysis. However, six out of nine patients that were thought to have died due to treatment-related toxicity were treated with dose-escalated radiotherapy while three had standard dose radiotherapy.

To investigate how many patients had an overall positive effect of the treatment, without recurrence, death, or serious late side effect, a composite outcome analysis was performed. This analysis showed no statistically significant difference between the two treatment groups, and more than 60% of patients were alive and had no recurrence or serious late side effect at 5 years after the end of radiotherapy. Similar analyses have been conducted in re-irradiation where the number of successful cases inevitably is much lower [60], but to our best knowledge this has not previously been explored in the primary-treatment setting. This analysis shows that the majority of patients treated in this study were free of disease and severe late side effects 5 years after treatment, regardless of having been treated with dose-escalated or standard dose radiotherapy.

Although we saw no statistically significant difference between the two treatment groups in the pattern of recurrence, there were more local recurrencies in the dose-escalated group than in the group treated with a standard dose. The imbalance in the T-stage distribution between the groups, where there is a predominance of smaller primary tumours (80% T1 and T2 tumours) in the standard dose group and a predominance of advanced stage primary tumours (55% T3 and T4 tumours) in the dose-escalated group, is expected to favour this pattern. Several previous studies have shown that advanced T-stage is a risk factor of recurrency [10,21,22]. The reason for the imbalance in T-stage distribution is probably that the decision to treat with either standard dose radiotherapy or dose-escalated radiotherapy was by the clinician’s choice based on local guidelines, and the guidelines primarily advocate dose-escalated radiotherapy to large primary tumours. It is worth noting that most patients affected by local recurrence in the dose-escalated cohort had HPV-positive tumours (Appendix A). This is especially interesting as there are several ongoing studies exploring the possibility of dose de-escalation in patients with HPV-positive oropharyngeal cancer. Published studies on de-escalation mainly include low-risk oropharyngeal cancer and often patients with T4 tumours are excluded, but some phase 2 trials including patients with T3 tumours have shown that these patients do not seem to have inferior outcomes when treated with de-escalated radiotherapy [61,62]. However, patients with T3 tumours make up a very small proportion of these studies and results from such sub-analyses should be treated with caution. The findings of our study, with local recurrencies in HPV-positive oropharyngeal tumours despite dose escalation, put the strategy of de-escalation in HPV-positive oropharyngeal cancer into question.

The variables chosen for matching the two treatment groups were tumour stage, concurrent medical treatment, and which data source the patients were included from (local quality registry or treatment planning system). We matched on tumour stage as it has an impact on survival [10], and although this variable was matched on, we still see a significant difference in the distribution of tumour stage. In the interest of cohort size, this imbalance was considered an acceptable compromise. Considering the aim to evaluate side effects, we matched on concurrent medical treatment, as different medical treatments have different toxicity profiles. We also matched on which data source the patients were included from, because the patients not identified by our local quality registry most often were patients who failed to come to regular follow-up visits. It was important to match on this factor since these patients might have died or been referred to hospice before the first scheduled follow-up visit and thus shared a negative prognostic profile. We chose not to match on more variables as this would further reduce the number of patients in each treatment group.

Previous studies on dose escalation in HNC have reported side effects on par with the side effects after standard dose radiotherapy. In contrast, the current study shows an increase of certain late side effects. This indicates that future studies on dose-escalated radiotherapy should preferably have a long follow-up in order to capture the long-term late side effects of the treatment, and also include studies on quality of life. Furthermore, future studies on finding predictive factors to select the subgroup of patients that might benefit from radiation dose escalation should preferably be set up as prospective randomised trials to avoid a biased control group using observational data. As the current study showed that the size of the primary tumour (GTVT) was a stronger prognostic factor than T-stage, this could be a clinical factor to evaluate in future studies exploring predictive factors in dose-escalation in HNC.

## 5. Conclusions

Patients treated with dose-escalated radiotherapy had a higher incidence of grade ≥3 ORN and late dysphagia compared to patients treated with standard dose radiotherapy. In the current study, we did not find any predictive factors to help select patients that would benefit from dose-escalated radiotherapy. However, the good OS in the dose-escalated cohort, despite an accumulation of poor prognostic factors, in combination with the recurrencies observed even among HPV-positive patients, encourages further attempts to identify such factors.

## Figures and Tables

**Figure 1 cancers-15-02580-f001:**
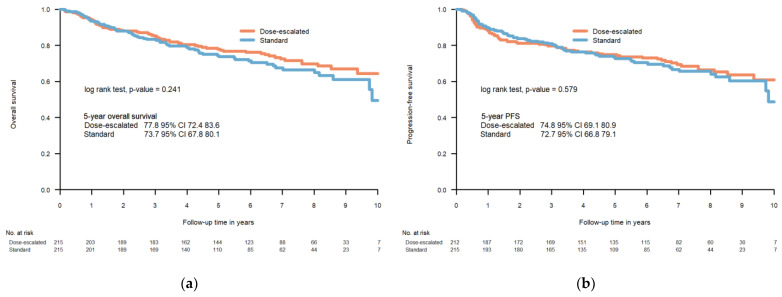
Overall survival (**a**) and progression-free survival (**b**) dose-escalated cohort vs. standard dose cohort.

**Figure 2 cancers-15-02580-f002:**
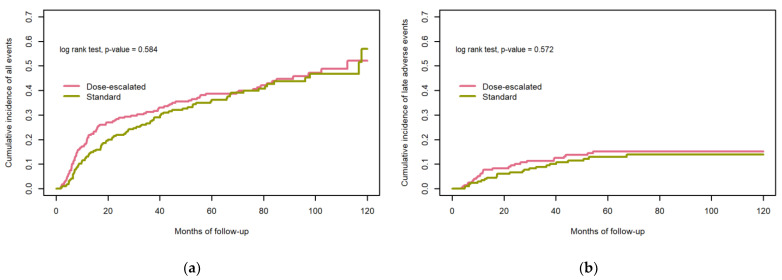
Composite outcome of recurrence, death, or late grade ≥3 toxicity in dose-escalated cohort vs. standard dose cohort (**a**), time to late grade ≥3 toxicity in dose-escalated cohort vs. standard dose cohort (**b**).

**Table 1 cancers-15-02580-t001:** Clinical characteristics and comparison between cohorts.

	Dose-Escalated	Standard Dose	*p*-Value	Overall
	Number (%)			Number (%)
**Subjects**	215	215		430
**Gender**			0.196	
Female	53 (24.7)	66 (30.7)		119 (27.7)
Male	162 (75.3)	149 (69.3)		311 (72.3)
**Age, mean (SD)**	62.2 (8.8)	64.5 (9.8)	0.011	63.3 (9.4)
**HPV status**			0.441	
Negative	33 (15.3)	40 (19.6)		73 (17.0)
Positive	182 (84.7)	175 (81.4)		357 (83.0)
**Performance status (PS)**			0.894	
PS 0	183 (85.1)	181 (84.2)		364 (84.7)
PS 1–2	32 (14.9)	34 (15.8)		66 (15.3)
**Tumour**			<0.001	
Base of tongue	120 (57.4)	33 (15.3)		153 (36.1)
Tonsil	89 (42.6)	166 (77.2)		255 (60.1)
Other *	0 (0.0)	16 (7.4)		16 (3.8)
**Smoking status**			0.789	
Never	70 (32.6)	65 (30.2)		135 (31.4)
Current	50 (23.3)	48 (22.3)		98 (22.8)
Former	95 (44.2)	102 (47.4)		197 (45.8)
**Tumour stage** †			0.012	
I-II	17 (7.9)	15 (7.0)		32 (7.4)
III	18 (8.4)	39 (18.1)		57 (13.3)
IVA-B	180 (83.7)	161 (74.9)		341 (79.3)
**T-stage**			<0.001	
T1	29 (13.5)	71 (33.0)		100 (23.3)
T2	67 (31.2)	101 (47.0)		168 (39.1)
T3	56 (26.0)	28 (13.0)		84 (19.5)
T4	63 (29.3)	15 (7.0)		78 (18.1)
**Therapy**			0.056	
Cetuximab	96 (44.7)	102 (47.4)		198 (46.1)
Cisplatin	65 (30.2)	64 (29.8)		129 (30.0)
Cisplatin + cetuximab	7 (3.3)	0 (0.0)		7 (1.6)
None	47 (21.9)	49 (22.8)		96 (22.3)
**Data source**			1.000	
Local quality registry	190 (88.4)	189 (87.9)		379 (88.1)
Dose planning system	25 (11.6)	26 (12.1)		51 (11.9)
**Recurrence**			0.073	
No recurrence	170 (79.1)	184 (85.6)		354 (82.3)
Local recurrence	22 (10.2)	10 (4.7)		32 (7.4)
Regional recurrence	4 (1.9)	5 (2.3)		9 (2.1)
Distant metastases	10 (4.7)	13 (6.0)		23 (5.3)
Locoregional + distant metastases	6 (2.8)	2 (0.9)		8 (1.9)
Progressive disease during RT	0 (0.0)	1 (0.5)		1 (0.2)
Not assessable	3 (1.4)	0 (0.0)		3 (0.7)

* Five patients with soft palate cancer and 11 patients with oropharyngeal cancer not otherwise specified. † Tumour stage according to the American Joint Committee on Cancer’s (AJCC) Cancer-Staging Manual, 7th edition. SD—standard deviation, HPV—human papillomavirus, PS—performance status according to WHO, RT—radiotherapy.

**Table 2 cancers-15-02580-t002:** Cox regression of mortality, multivariable analysis.

Mortality			
	HR	95% CI	*p*-Value
**Cohort**			
Standard dose	1.00	reference	
Dose-escalated	0.97	0.68–1.40	0.885
**HPV status**			
Negative	1.00	reference	
Positive	0.45	0.30–0.68	<0.001
**ECOG PS**			
PS 0	1.00	reference	
PS 1–2	1.51	1.01–2.24	0.043
**T stage**			
T1-T2	1.00	reference	
T3-T4	1.42	0.93–2.16	0.108
**Smoking status**			
Never	1.00	reference	
Former	1.33	0.83–2.13	0.239
Current	2.57	1.57–4.23	<0.001
**Age (per year)**	1.04	1.02–1.06	<0.001
**Normalised Age ***	1.46	1.21–1.77	<0.001
**GTVT (** **per cm^3^)**	1.01	1.00–1.02	0.010
**Normalised GTVT †**	1.22	1.05–1.41	0.010

* Normalisation of the regression coefficient was performed by multiplication with the standard deviation of the age parameter in the cohort. † Normalisation of the regression coefficient was performed by multiplication with the standard deviation of the GTVT volumes parameter in the cohort. HPV—human papillomavirus, PS—performance status according to WHO, GTVT—volume of the primary tumour.

**Table 3 cancers-15-02580-t003:** Comparison of grade ≥3 side effects between treatment cohorts and in total cohort.

	Dose-Escalated		Standard Dose			Total	
	Number (%)		Number (%)		*p*-Value	Number (%)	
**Skin**					0.240		
Acute	51	(23.7)	62	(28.8)		113	(26.3)
Late	10	(4.7)	5	(2.3)		15	(3.5)
None	154	(71.6)	148	(68.8)		302	(70.2)
**Osteoradionecrosis**					0.001		
Late	19	(8.8)	4	(1.9)		23	(5.3)
None	196	(91.2)	211	(98.1)		407	(94.7)
**Larynx**					1.000		
Acute	0	(0.0)	0	(0.0)		0	(0.0)
Late	3	(1.4)	2	(0.9)		5	(1.2)
None	212	(98.6)	213	(99.1)		425	(98.8)
**Salivary glands**					0.229		
Late	15	(7.0)	22	(10.2)		37	(8.6)
None	199	(92.6)	193	(89.8)		392	(91.2)
**Trismus**					0.053		
Acute	1	(0.5)	2	(0.9)		3	(0.7)
Late	8	(3.7)	1	(0.5)		9	(2.1)
None	206	(95.8)	212	(98.6)		418	(97.2)
**Mucosa**					0.350		
Acute	135	(62.8)	128	(59.5)		263	(61.2)
Late	5	(2.3)	2	(0.9)		7	(1.6)
None	75	(34.9)	85	(39.5)		160	(37.2)
**Dysphagia**					0.012		
Late	39	(18.1)	21	(9.8)		60	(14.0)
None	176	(81.9)	194	(90.2)		370	(86.1)

**Table 4 cancers-15-02580-t004:** Grade 5 toxicity.

Patient	Age (Years)	Gender	Performance Status at Start of Radiotherapy	Tumour Stage *	Dose Level	Cause of Death	Time of Death (After End of Radiotherapy)
1	81	Male	0	IVA	Dose-escalated	Massive pharyngeal bleeding	6 months
2	79	Female	0	IVA	Dose-escalated	Acute radiation toxicity	3 weeks
3	73	Male	2	II	Dose-escalated	Acute radiation toxicity	6 days
4	70	Male	1	III	Dose-escalated	Infection	3 weeks
5	68	Male	0	IVA	Standard dose	Acute radiation toxicity	10 days
6	65	Male	0	IVA	Standard dose	Acute radiation toxicity	4 weeks
7	63	Male	0	IVA	Dose-escalated	ORN and infection	5 years
8	63	Female	0	IVA	Standard dose	Consequential late effects, infection	6 months
9	59	Male	0	IVA	Dose-escalated	Massive pharyngeal bleeding	1 year

* Tumour stage according to the American Joint Committee on Cancer’s (AJCC) Cancer-Staging Manual, 7th edition. ORN—osteoradionecrosis

## Data Availability

Research data are stored in an institutional repository and will be shared upon reasonable request to the corresponding author.

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
