# Peer review of "Dose Escalation of Oropharyngeal Cancer: Long-Time Follow-Up and Side Effects"

_cancers, 2023, doi:10.3390/cancers15092580_

Round 1
Reviewer 1 Report
Reviewers’ opinion: It is a well-written, well-documented manuscript dealing with a clinically relevant and important issue. The main values of the work are the elaboration of a great real-world database, the presentation of long-term treatment results with RT dose escalation in advanced head and neck cancers, the composite outcome data processing, and the true real-life information about the incidence of serious RT toxicities. My concerns/ notifications/ suggestions are the following:
Abstract: I feel important to report in the abstract as well that dose escalation means EBRT dose escalation or brachytherapy boost.
I do not understand the great difference (18 months) between the median follow-up times of the two treatment groups, since there was no significant difference between the survival times. (The difference between the longest follow-up times was 9 months as well.)
I consider as an important message that the survival effect of RT dose escalation in advanced tumors is excellent. It would be worthy to mention it in the abstract as well.
Lines 76-77.: Note: the one-physician’s decision maybe not the best decision, mainly in the indication of the brachytherapy boost.
Line 96.: The Propensity Score Matching was based on tumor stage, concurrent medication, and the data source.” Meanwhile there was a greater number of advanced tumors in the dose escalated cohort. Please clarify it.
Lines 135-136.: The weekly form of cisplatin administration was the standard local procedure in the institute in those times? What was the reason for maximizing the dose of cisplatin?
Table 1: Tumor Stage IVA-B.
Table 1.: There was a higher rate of local recurrence in the dose escalated group (see in Discussion section as well), however the survival data was slightly better in this cohort. Please clarify it.
Figures 1. and 2.: The graphs are not visible entirely.
Table 3.: The incidence of osteo-radionecrosis (ORN) in the dose escalated group was 8,8%. However, about the half of patients received brachytherapy boost (see TableA1) and considering this type of intervention the onset of late ORN is typically not so frequent. So, what was percentage of ORN in case of external beam dose escalation?
Lines 333-335.: “follow-up is longer in the dose-escalated group, which could have contributed to the higher rate of late toxicity” What was the general onset times of late toxicities?
Lines 378-379.: It is an interesting finding.
Reviewer 2 Report
This retrospective propensity score matching study compared survival and complications in oropharynx cancer patients treated with dose-escalated RT and standard dose RT with a long follow-up time. Because of long follow-up time, this study can provide valuable results. In addition, this manuscript is well written and concise.
However, very important prognostic factors including age, primary tumor site, AJCC stage, and T stage were imbalanced and skewed between two patient groups. So, reliable results and conclusions can not be drawn from this study. Significant prognostic factors should be evenly distributed between interventional and matched cohorts.
Reviewer 3 Report
The manuscript written by Embring A et al., is related to the association between two types of treatment (radiotherapy) in oropharyngeal cancer. the study made by the authors is interesting, well redacted, and well explained, this manuscript could provide in the future new tools for evaluating the modalities of RT treatment in HNC.
Despite the interest of this manuscript, I have some questions and suggestions that the authors need to consider.
First. 2.3 Toxicity outcomes
This variable is interesting; however, the authors did not evaluate the oral mucositis as a side effect in the treatment of HNC. I suggest including the oral mucositis in their study.
Second. Figure 1
Please check figure 1. That one is incomplete (b) and impossible to review.
Third. Table 2
Would it be possible to add the sex (female and male) as a variable and compare it with mortality?
Fourth. Table 3.
In the variable mucosa, why didn't you use grades of mucositis according to the WHO (Grades I to III) instead of Acute-None?
Fifth. Discussion
Lines 317 to 318 "In the current..." Why, in this paragraph, do not you add the variable mucositis and discuss it with grades of dysphagia and ORN and explain any possible relationship between those variables?
Lines 345 to 348 "In the current study..." Would it be interesting to discuss mucositis between groups and these variables? Would it be possible to do it?
Line 350 "Nine patients..." This line is interesting, but why did you not discuss it in more detail? For example, "Nine patients... the majority of them had Clinical Stage III and ORN with oral mucositis, and there were deaths due to...
Line 379 "(supplementary material..." I suggest instead of using "appendix", use supplementary tables (S1, S2, etc.).
Table A2. I did not see any relationship between table A2 and the discussion. Why use Table A2 in the discussion section? Why did you not describe it in the results section, and why did you not name it as Table A1?
Fifth: Appendix A
Line 441 "We compared..." Why did you not evaluate DFS (disease-free survival) and compare it with OS and PFS?
Table A2 I suggest the table A2 have any relationship with the discussion or use this table in the results section and describe it adequately; why is it not named S2?
Please add the number of approvals from the National Ethical Review Authority.
Overall comments
This manuscript is interesting since the authors study and compare modalities of treatment, but it contains some mistakes that make it confusing.
Reviewer 4 Report
the paper is comparing the conventional and dose escalated radiotherapy for oropharyngeal cancer patients
The publication sounds scientific and beneficial to the field however needs some correction:
1. figures quality is low.
2. what is the novelty? is it not predictable to have more side effects after high dose radiation?
3. introduction is too short! write a paragraph about the side effects that you studied and the definition of escalated radiotherapy
4. is escalated radiotherapy the same as palliative radiotherapy since many patients were end-stage? please explain it
5. add a paragraph showing potential studies in the field or future studies regarding escalated radiotherapy
Round 2
Reviewer 1 Report
I accept the answers and corrections of the authors and I think that the quality of the material has improved after the corrections.
Reviewer 2 Report
The manuscript is not modified as I previously asked.
Reviewer 3 Report
The manuscript was improved compared with the previous version.
Reviewer 4 Report
The corrections are acceptable.